# CMYA5 establishes cardiac dyad architecture and positioning

Fujian Lu[1], Qing Ma[1], Wenjun Xie[2], Carter L. Liou[1], Donghui Zhang[1,3], Mason E. Sweat[1], Blake D. Jardin [1], Francisco J. Naya[4], Yuxuan Guo[1,5], Heping Cheng [6] & William T. Pu[1,7 ✉]

Cardiac excitation-contraction coupling requires dyads, the nanoscopic microdomains formed adjacent to Z-lines by apposition of transverse tubules and junctional sarcoplasmic reticulum. Disruption of dyad architecture and function are common features of diseased cardiomyocytes. However, little is known about the mechanisms that modulate dyad organization during cardiac development, homeostasis, and disease. Here, we use proximity proteomics in intact, living hearts to identify proteins enriched near dyads. Among these proteins is CMYA5, an under-studied striated muscle protein that co-localizes with Z-lines, junctional sarcoplasmic reticulum proteins, and transverse tubules in mature cardiomyocytes. During cardiac development, CMYA5 positioning adjacent to Z-lines precedes junctional sarcoplasmic reticulum positioning or transverse tubule formation. CMYA5 ablation disrupts dyad architecture, dyad positioning at Z-lines, and junctional sarcoplasmic reticulum $Ca^{2+}$ release, leading to cardiac dysfunction and inability to tolerate pressure overload. These data provide mechanistic insights into cardiomyopathy pathogenesis by demonstrating that CMYA5 anchors junctional sarcoplasmic reticulum to Z-lines, establishes dyad architecture, and regulates dyad $Ca^{2+}$ release.

[1] Department of Cardiology, Boston Children's Hospital, 300 Longwood Avenue, Boston, MA 02115, USA. [2] The Key Laboratory of Biomedical Information Engineering of Ministry of Education, Institute of Health and Rehabilitation Science, School of Life Science and Technology, Xi'an Jiaotong University, 710049 Xi'an, Shanxi, China. [3] State Key Laboratory of Biocatalysis and Enzyme Engineering, School of Life Science, Hubei University, 430062 Wuhan, Hubei, China. [4] Department of Biology, Program in Cell and Molecular Biology, Boston University, Boston, MA 02215, USA. [5] Peking University Health Science Center, School of Basic Medical Sciences, The Institute of Cardiovascular Sciences, Key Laboratory of Molecular Cardiovascular Science of Ministry of Education, 100191 Beijing, China. [6] State Key Laboratory of Membrane Biology, Institute of Molecular Medicine, Peking-Tsinghua Center for Life Sciences, Peking University, 100871 Beijing, China. [7] Harvard Stem Cell Institute, 7 Divinity Avenue, Cambridge, MA 02138, USA. ✉email: william.pu@cardio.chboston.org

Cardiomyocytes exemplify the integration of form and function, in which the precise positioning of nanoscale structures enables efficient, coordinated cycles of contraction and relaxation[1]. Sarcomeres, concatenated end-to-end at structures known as Z-lines to form myofibrils, drive cardiomyocyte contraction. Forceful cardiomyocyte contraction requires coordinated activation of individual sarcomeres throughout the cell. This is achieved through excitation–contraction (E-C) coupling. The initiating signal for cardiomyocyte contraction is the action potential, which propagates rapidly along the plasma membrane. In mature cardiomyocytes, the plasma membrane forms highly ordered tubular invaginations, T-tubules, that penetrate into the interior of cardiomyocytes and allow rapid transmission of the action potential throughout the cardiomyocyte interior[2]. At structures termed dyads, T-tubules are juxtaposed to the junctional sarcoplasmic reticulum (jSR), a specialization of the sarcoplasmic reticulum (SR)[1,3]. The L-type $Ca^{2+}$ channel, housed within T-tubule membranes, opens upon membrane depolarization and releases extracellular $Ca^{2+}$ into the dyadic cleft or junction, the narrow (~12 nm) cytoplasmic space between the T-tubules and the jSR. This localized increase in $Ca^{2+}$ activates RYR2, the major intracellular $Ca^{2+}$ release channel of cardiomyocytes. Located on the jSR, activated RYR2 releases $Ca^{2+}$ from the jSR lumen into the dyadic cleft, from which it subsequently diffuses to stimulate contraction of adjacent sarcomeres. Because of the relatively slow speed of diffusion, the positioning of dyads adjacent to Z-lines is important for coordinated contraction[4].

How sarcomeres, jSR, and T-tubules attain their intricate nanoscale organization within the cardiomyocyte and with respect to each other is a fundamental mystery in cardiac biology. Studies have focused on factors that regulate T-tubule formation and organization. cBIN1 (cardiac bridging integrator 1), a membrane-binding protein[5], JPH2 (junctophilin 2), a protein that interacts with both the cell membrane and jSR[6], and RYR2[7] have been found to be essential for the formation or maintenance of T-tubules. However, factors responsible for jSR organization and positioning have not been identified.

Disruption of dyad architecture and impaired E-C coupling are common features of failing cardiomyocytes[8]. In rodent pressure overload, T-tubule disorganization preceded ventricular dysfunction[9]. T-tubules were less closely associated with jSR, a lower fraction of Z-lines had associated dyads, and the distance from dyads to Z-lines increased[4]. These changes were associated with impaired E-C coupling[4]. Similar changes to T-tubules, jSR, and dyads were found in human dilated and ischemic cardiomyopathy[10].

CMYA5 (cardiomyopathy-associated protein 5), also known as myospryn, is an under-studied ~450 kDa protein that is selectively expressed in cardiac and skeletal muscle[11,12]. CMYA5 is a member of the tripartite motif-containing super-family (TRIM), which contain four protein-protein interaction domains (RING, BBox1, BBox2, and coiled-coiled) in a conserved order and generally functions as part of large protein complexes[13]. Based on its co-expression with known cardiomyopathy genes, *CMYA5* initially was linked hypothetically to cardiomyopathy[14]. This link gained empirical support when a *CMYA5* coding single nucleotide polymorphism was associated with left ventricular wall thickness and diastolic dysfunction[15]. CMYA5 was previously reported to interact with multiple muscle proteins, including RYR2[12], the Z-line protein ACTN2 (α-actinin2)[11], desmin[16], titin[17], and PKA (protein kinase A)[18]. However, little has been reported about the in vivo function of CMYA5. A recent study published while this manuscript was in preparation demonstrated that CMYA5 knockout causes cardiac dysfunction and mislocalization of RYR2[19]. However the effect of CMYA5 knockout on dyad formation, structure, and function was not investigated in detail.

Here we performed a proximity proteomics screen for components of dyads and identified CMYA5. We found that CMYA5 is required to efficiently position jSR adjacent to Z-lines, an early and essential step in dyad assembly. Ablation of CMYA5 disrupted dyad architecture, dysregulated RYR2 channel activity, and impaired the fidelity of E-C coupling. Finally, CMYA5 protected heart function and dyad structure from pressure overload stress.

## Results

**Proximity proteomics identifies CMYA5 as a dyadic protein.** We developed a system to identify proteins that localize in the vicinity of dyads using BioID, a proximity labeling technique with an estimated labeling radius of ~10 nm[20,21], which matches the approximate diameter of the dyadic cleft. We fused BirA*[20], an enzyme that releases short-lived biotin-free radicals, to the N-terminus of either junctin (gene symbol: ASPH; BirA*-ASPH) or triadin (BirA*-TRDN), transmembrane jSR proteins that form a complex with RYR2[3,22]. By localizing BirA* to the dyadic cleft, we anticipated that dyadic proteins would become biotinylated, so that they could be purified on streptavidin and subsequently identified by mass spectrometry. We expressed these dyad-targeted proteome biosensors in vivo in murine hearts using adeno-associated virus serotype 9 (AAV9) and the cardiac troponin T (*Tnnt2*) promoter (Fig. 1a). We administered AAV to P1 mice, supplied biotin from P21 to P28, and collected hearts at P28 (Fig. 1b). Histological analysis confirmed efficient AAV transduction of cardiomyocytes (70 and 78% for BirA*-ASPH and BirA*-TRDN, respectively; Fig. 1c) and punctate staining of the BioID sensors overlying T-tubules, marked by CAV3 (caveolin 3) (Fig. 1d), consistent with appropriate expression and localization of the biosensors to dyads without perturbation of their overall organization. Probing protein lysates with streptavidin conjugated to horseradish peroxidase confirmed biotin- and BirA*-dependent biotinylation of protein complexes (Fig. 1e). Biotinylated proteins were purified using immobilized streptavidin and analyzed by mass spectrometry (Fig. 1f). We ranked genes by the ratio of the average protein signal in the BirA*-ASPH and BirA*-TRDN groups to that in the control group. Among the proteins highly enriched in the BioID groups were RYR2, JPH2, ASPH, and TRDN (Fig. 1f and Supplementary Data 1). Recovery of these known dyadic proteins validated our experimental strategy.

We prioritized proteins found in both BioID groups that lacked signal in the control group. Among these proteins, one of the most highly enriched in the BioID groups was CMYA5, a protein expressed in striated muscle and neurons[23,24]. Within the heart, CMYA5 is selectively expressed in cardiomyocytes (Supplementary Fig. 1)[25]. CMYA5 has been reported to interact with multiple cardiomyocyte proteins (e.g., RYR2[12], ACTN2[11], desmin[16], titin[17], and PKA.[18]) that have distinct localization patterns. In mature cardiomyocytes, we confirmed that CMYA5 co-localizes with jSR protein RYR2 in a striated pattern, with periodic peaks every 2 μm (Fig. 1g). We further tested the in situ proximity of CMYA5 and RYR2 using the proximity ligation assay (PLA), which detects proteins in close proximity (< 40 nm)[26]. Consistent with the reported interaction of CMYA5 with RYR2[12], we observed bright, punctate PLA signals in a striated pattern throughout adult cardiomyocytes, marking loci where CMYA5 and RYR2 are in close proximity (Fig. 1h). These results indicate that CMYA5 is a dyadic protein that is closely associated with RYR2.

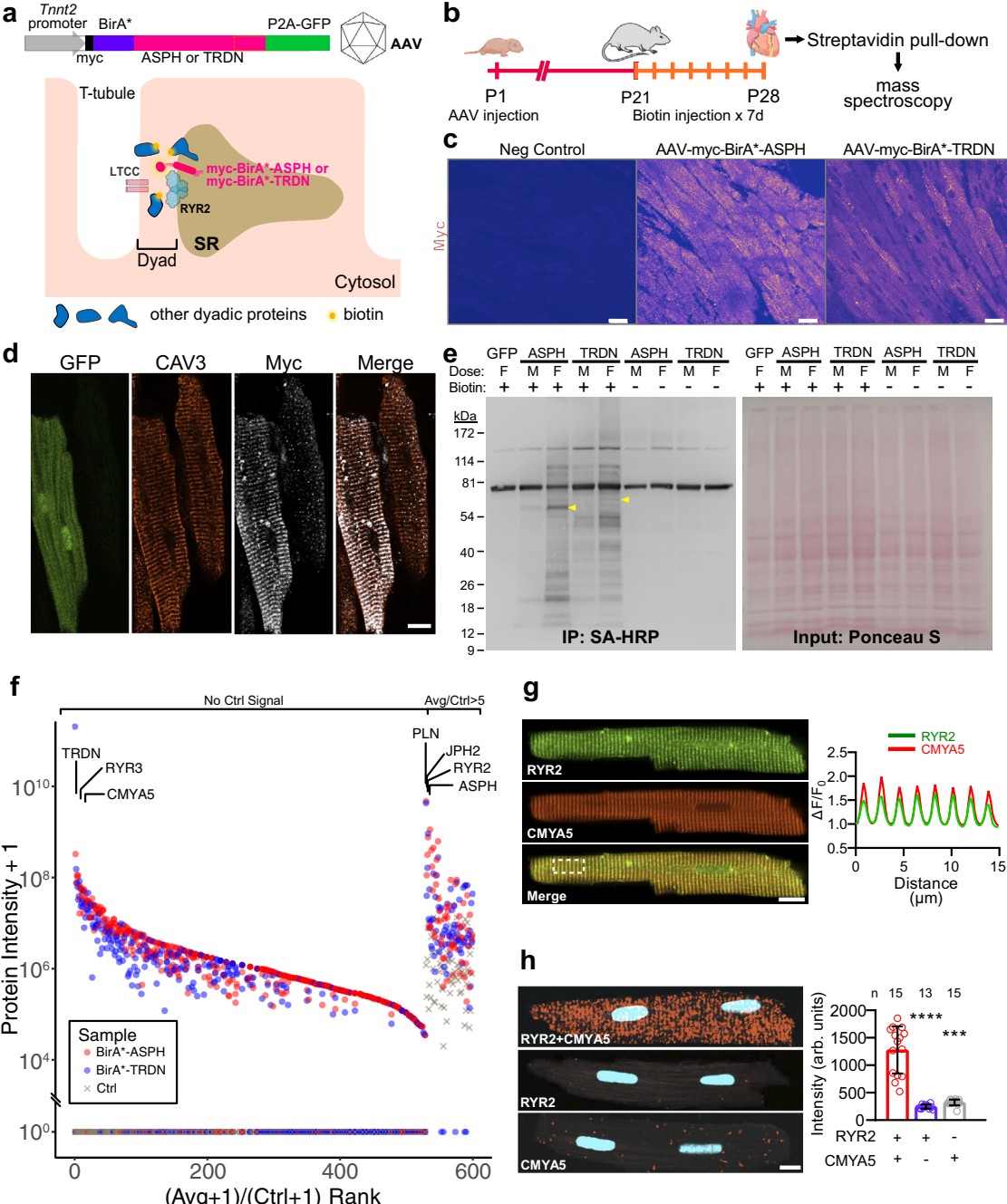

**Fig. 1 BioID identification of dyadic proteins. a** Proximity proteomics strategy. BirA* and myc epitopes were fused to ASPH and TRDN, which interact with RYR2 in jSR of dyads. N-terminal fusion positions BirA* in the dyadic cleft. The fusion protein was expressed in vivo in cardiomyocytes using AAV9 and the cardiomyocyte-selective troponin T (Tnnt2) promoter. GFP was co-expressed using a P2A sequence. **b** Experimental timeline. Artwork from biorender.com. **c** Expression of BirA* dyadic biosensors in myocardium. Heart sections were stained for the myc epitope tag. Most cardiomyocytes were immunoreactive. Bar = 20 μm. Representative of 3 independent experiments. **d** Mature ventricular cardiomyocytes. In the left GFP + cell, myc immunoreactive signal co-localized with CAV3, a T-tubule marker. Bar = 10 μm. Representative of three independent experiments. **e** Protein lysates from mice expressing the indicated BirA*-fused biosensors or GFPs were analyzed by western blotting. AAV dose and treatment with exogenous biotin are indicated. M, middle dose ($2 \times 10^{10}$ vg/g). F, full dose ($5.5 \times 10^{10}$ vg/g). Ponceau S, total protein. SA-HRP labeled biotinylated proteins. Representative of two independent experiments. **f** Summary of mass spectrometry data. Red and blue symbols indicate protein signals from BirA* biosensors. Gray symbols show control (AAV-GFP) signals. One replicate pooled from three hearts was performed for each sensor and for control. Proteins were ranked by the ratio of the average signal of the two different BirA* biosensors to the control signal. Proteins lacking control signal grouped to the left. Only proteins with average/control ratio >5 are shown. **g** Co-localization of CMYA5 and jSR marker RYR2. Immunostained adult cardiomyocytes were imaged with a confocal microscope. Spatial profiles to the right demonstrate co-localization in a striated pattern with 2 μm spacing. Bar = 10 μm. **h** Proximity ligation assay. Punctate red signal, indicating close proximity of RYR2 and CMYA5, was observed when both RYR2 and CMYA5 antibodies were included, but not with either in isolation. Signal intensity is quantified at the right. *n*, number of cardiomyocytes. Two-sided Kruskal–Wallis with Dunn's multiple comparison test vs. RYR2 + CMYA5: ***, $P < 0.001$; ****, $P < 0.0001$. Data are presented as mean ± SD. Bar = 10 μm.

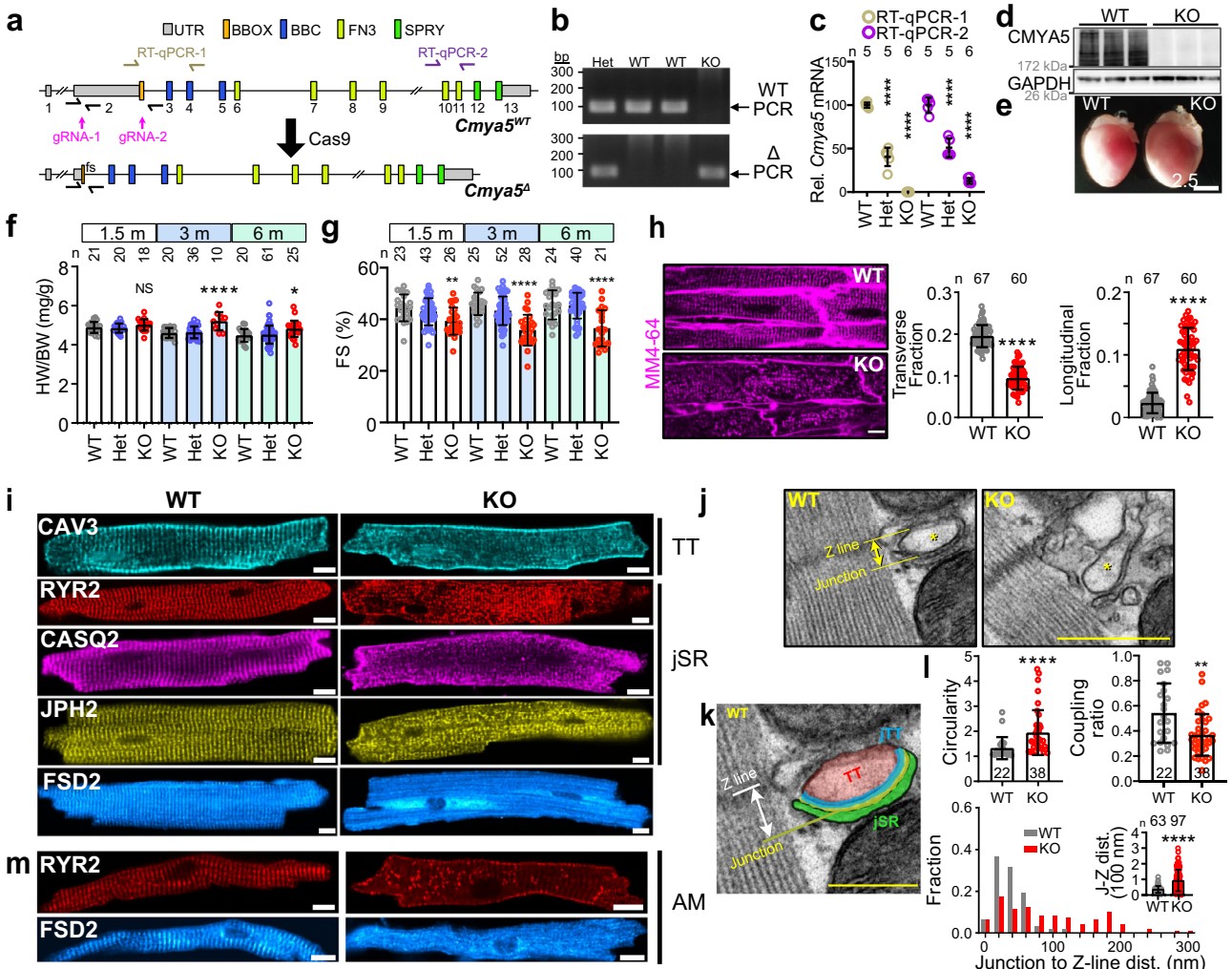

**Fig. 2 Characterization of hearts lacking CMYA5. a** Genomic structure of *Cmya5* wild-type and Δ alleles. Cas9-mediated deletion of 9281 bp of exon 2 causes a frameshift (fs) after the 55th amino acid residue. qRTPCR and genotyping amplicons are indicated. **b** PCR genotyping using WT (1 + 2) and Δ (1 + 3) primers. Representative of five independent experiments. **c** *Cmya5* cardiac mRNA levels. RT-qPCR amplicon 1 and 2 represent the deleted region and the 3′ end of the transcript, respectively. ANOVA with Dunnett's multiple comparison test vs. WT of the same amplicon. *n*, number of hearts. Data are presented as mean ± SD. **d** Cardiac protein lysates were analyzed by western blotting. KO samples lacked CMYA5 immunoreactivity. Representative of five independent experiments. **e** Gross morphology of WT and KO hearts. Bar = 2.5 mm. Representative of three independent experiments. **f** Heart weight normalized to body weight, at the indicated ages. ANOVA with Dunnett's multiple comparison test vs. WT at the same time point. *n*, number of hearts. **g** Echocardiographic measurement of systolic heart function. FS fractional shortening. *n*, sample size. Kruskal–Wallis with Dunn's multiple comparison test vs. WT at the same time point. **h** In situ T-tubule imaging. After plasma membrane labeling by MM4-64. 3-month-old hearts were optically sectioned using a confocal microscope. Right, transverse, and longitudinal T-tubule fractions. Mann–Whitney. Bar = 10 μm. *n*, number of cells. **i** Isolated ventricular WT or KO cardiomyocytes immunostained for T-tubule (TT: CAV3) and jSR (RYR2, CASQ2, JPH2) markers, as well as FSD2. Bar = 10 μm. **j–l** Transmission electron microscopy of WT or KO ventricular myocardium. *, T-tubule. The WT T-tubule micrograph is enlarged and labeled in **k**. **l** Quantification of T-tubule parameters, defined in Supplementary Fig. 4a. *n*, number of dyads from at least 10 cardiomyocytes from 3 different mice. Mann–Whitney. Bar = 500 nm. **m** Isolated WT or KO atrial cardiomyocytes immunostained for RYR2 or FSD2. NS, not significant; *, *P* < 0.05; **, *P* < 0.01; ***, *P* < 0.001; ****, *P* < 0.0001. Statistical tests were two-sided. Data are presented as mean ± SD.

**CMYA5 is required for normal heart function.** The in vivo function of CMYA5 has only been described in one study, which found that its ablation causes mild cardiac dysfunction, mislocalization of RYR2, and ultrastructural abnormalities of the SR and T-tubules[19]. However, the contribution of CMYA5 to dyad development, organization, and function was not investigated. We obtained an independent *Cmya5* loss-of-function mouse model in which CRISPR/Cas9 and two guide RNAs (gRNAs) were used to delete 9281 bp in exon 2 (Fig. 2a; "Methods"). The mutant allele, *Cmya5*Δ, is predicted to cause a frameshift mutation after amino acid residue 55. We confirmed the *Cmya5* genomic deletion (Fig. 2b) and its resultant

depletion of *Cmya5* transcript (Fig. 2c) and protein (Fig. 2d). Gross examination of *Cmya5*Δ/Δ (KO) hearts revealed substantial cardiac remodeling, including an increase in heart size (Fig. 2e) and weight, normalized to body weight, at 3 months of age (Fig. 2f). Echocardiographic analysis revealed diminished systolic ventricular function in KO mice, consistent with a recent study of an independently generated null allele[19]. Heterozygotes (Het) had normal heart weight and ventricular function (Fig. 2f, g). Diminished systolic function was accompanied by ventricular dilatation (Supplementary Fig. 2). These data indicate that *Cmya5* is essential for normal heart function.

**CMYA5 establishes normal dyad architecture and positioning adjacent to Z-lines**. The architecture of cardiac dyads is critical for efficient E-C coupling. To assess the effect of CMYA5 ablation on dyad organization and structure, we first examined the T-tubule system by optically sectioning hearts stained with the plasma membrane dye MM 4-64[27]. In KO cardiomyocytes, we observed dramatic disruption of T-tubules (Fig. 2h). Quantitative image analysis[9] confirmed significantly reduced transverse fraction and increased longitudinal fraction in KO compared to control (Fig. 2h). Next, we imaged T-tubules and jSR by immunostaining component proteins in isolated adult cardiomyocytes. Both T-tubule (CAV3) and jSR (RYR2, CASQ2 (calsequestrin 2), JPH2) markers were disorganized in KO (Fig. 2i). FSD2 (fibronectin type III and SPRY domain containing 2), a protein closely related to the C-terminus of CMYA5[12] (Supplementary Fig. 3a), also lost its jSR localization pattern[12] (Fig. 2i). To confirm that CMYA5 ablation cell autonomously disrupts T-tubule organization in the absence of cardiac dysfunction, we performed mosaic *Cmya5* inactivation using Cas9 and AAV-mediated somatic mutagenesis[7] (CASAAV; Supplementary Fig. 3b). By administering low-dose AAV-gRNA[*Cmya5*], we ablated *Cmya5* in a minority of cardiomyocytes without impairing heart systolic function (Supplementary Fig. 3c, d). Consistent with the *Cmya5*$^{\Delta/\Delta}$ results, the CASAAV-transduced cardiomyocytes, marked by GFP, exhibited T-tubule disorganization (Supplementary Fig. 3e–g). Together, these data confirm that CMYA5 is cell autonomously required for T-tubule organization and underscore the importance of CMYA5 for normal cardiac function.

To evaluate dyad architecture with greater resolution, we analyzed cardiomyocytes by transmission electron microscopy (TEM). T-tubules, jSR, and their intimate relationship with each other were all perturbed in KO (Fig. 2j, k and Supplementary Fig. 4a). These abnormalities were quantified by measuring T-tubule circularity, T-tubule-SR coupling ratio (the fraction of each T-tubule juxtaposed to jSR), and Junction-Z-line distance (perpendicular distance from Z-line to jSR-T-tubule junctions), as defined in Supplementary Fig. 4a and Zhang et al.[10]. KO T-tubules were less circular, had reduced spatial coupling, and had greater and more heterogeneous Junction-Z-line distance (Fig. 2l). Together, these changes to dyad architecture and positioning would be expected to make E-C coupling weaker and more variable. While jSR morphology was abnormal in KO, overall Z-line and endoplasmic reticulum (ER) organization were unaffected, as revealed by the immunofluorescent patterns of SR protein SERCA2a (sarco/endoplasmic reticulum Ca$^{2+}$-ATPase 2a) and Z-line protein ACTN2 and by AAV-mediated expression of an ER-targeted fluorescent protein (Supplementary Fig. 4b–e). Although CMYA5 has been reported to interact with PKA and titin, we did not observe altered localization of PKA (Supplementary Fig. 5a) or titin (Supplementary Fig. 5b). Moreover, measurement of nuclear and cytoplasmic NFAT3 did not support the activation of the hypertrophic calcineurin pathway (Supplementary Fig. 5c, d), which is downstream of calcineurin. Although *Cmya5* KO was reported to cause abnormal cardiac mitochondria[19], we did not detect abnormalities in the overall mitochondrial organization or mitochondrial morphology (Supplementary Fig. e–g).

Collectively, these data indicate that CMYA5 is required for dyad architecture and positioning with respect to Z-lines.

**CMYA5 regulates the hierarchical assembly of dyads**. To further dissect the inter-relationships between T-tubules, jSR, and Z-lines in normal and *Cmya5* KO cardiomyocytes, we analyzed adult murine atrial cardiomyocytes, which have a paucity of T-tubules[28]. RYR2-containing SR domains can be found just under the sarcolemma (peripheral couplings) or localized adjacent to Z-lines (corbular SR), as demonstrated in Fig. 2m and Supplementary Fig. 4f[29]. These observations indicate that RYR2 positioning adjacent to Z-lines does not require T-tubules. In *Cmya5* KO atrial cardiomyocytes, RYR2, FSD2, and CASQ2 lost their characteristic co-localization with Z-lines (Fig. 2m and Supplementary Fig. 6a), demonstrating that CMYA5 is also required in atrial cardiomyocytes to tether corbular SR to Z-lines. Notably, RYR2 and JPH2 localization in peripheral couplings near the sarcolemma was preserved in KO atrial cardiomyocytes (Fig. 2m and Supplementary Fig. 6a), indicating a selective role for CMYA5 localization of the subset of RYR2-containing SR domains normally positioned adjacent to Z-lines.

We further examined the organization of Z-lines, jSR, and T-tubules in control and KO ventricular cardiomyocytes at different developmental stages. In P7 cardiomyocytes, T-tubules, marked by CAV3, have not yet organized[30], but RYR2, CMYA5, and FSD2 were already co-localized in a striated pattern that corresponds to sarcomere Z-lines (Fig. 3a–c). CMYA5 ablation, however, disrupted the Z-line distribution of RYR2 (Fig. 3c). These data further demonstrate that CMYA5 is required and T-tubules dispensable for the positioning of RYR2 and jSR to Z-lines. CMYA5 deficiency resulted in reduced levels of jSR proteins RYR2, FSD2, and cBIN1 (Supplementary Fig. 6b), likely due to their degradation because of impaired jSR localization and assembly. We gained further insights by examining E15.5 cardiomyocytes, whose actively assembling subcellular structures are in different states of organization. Confocal imaging of *Ryr2-GFP*[31] knockin heart sections revealed subsets of Z-lines co-localized with both CMYA5 and RYR2 (yellow arrowheads, Fig. 3d and Supplementary Fig. 6c), CMYA5 only (red arrowheads, Fig. 3d and Supplementary Fig. 6c), or neither CMYA5 nor RYR2 (white arrowheads, Fig. 3d and Supplementary Fig. 6c). Z-lines that co-localized with RYR2 but not CMYA5 were rare. Together, these data show that CMYA5 localization at Z-lines does not require co-localization of jSR/RYR2 or T-tubules, whereas efficient T-tubule and jSR/RYR2 localization at Z-lines does require CMYA5.

To further test this organizational hierarchy, in which CMYA5 reads Z-line positional information to properly localize jSR and T-tubules, we used CASAAV[7] to perform mosaic ablation of sarcomeres or RYR2. RYR2 mosaic somatic deletion by CASAAV[7] (Supplementary Fig. 6d) did not perturb the normal striated staining pattern of CMYA5 and FSD2 (Fig. 3e, f). In contrast, CASAAV-mediated mosaic depletion of sarcomere protein MYH6[32] (Supplementary Fig. 6e) disrupted the organization of Z-lines, CMYA5, and jSR, marked by JPH2 (Fig. 3g, h and Supplementary Fig. 6f–i). Taken together, these results demonstrate that dyads are built on scaffolding provided by sarcomeres. CMYA5 localizes to Z-lines, and subsequently contributes to tethering jSR adjacent to these structures. T-tubules subsequently form and co-localize with jSR, yielding organized, properly positioned dyads.

To investigate the features of CMYA5 required for its activity, we adopted a rescue strategy in which we used AAV to express CMYA5-related protein fragments. At 11.2 kb, the *Cmya5* coding region exceeds the cargo capacity of AAV. All known protein-protein interactions involving CMYA5 map to its C-terminal end[12,33], contained within a fragment named MD9 (amino acids 2731–3739) that was found to bind and cluster RYR2 in a heterologous expression system (Supplementary Fig. 3a)[12]. Furthermore, FSD2 shares 45.5% protein sequence similarity with the C-terminus of CMYA5 and retains the same domain architecture and order (Supplementary Fig. 3a)[12]. For these reasons, we assayed CMYA5$^{MD9}$ and FSD2 for their ability to rescue *Cmya5* KO hearts. CMYA5$^{MD9}$ and FSD2 expressed from AAV vectors localized in a striated pattern in WT cardiomyocytes

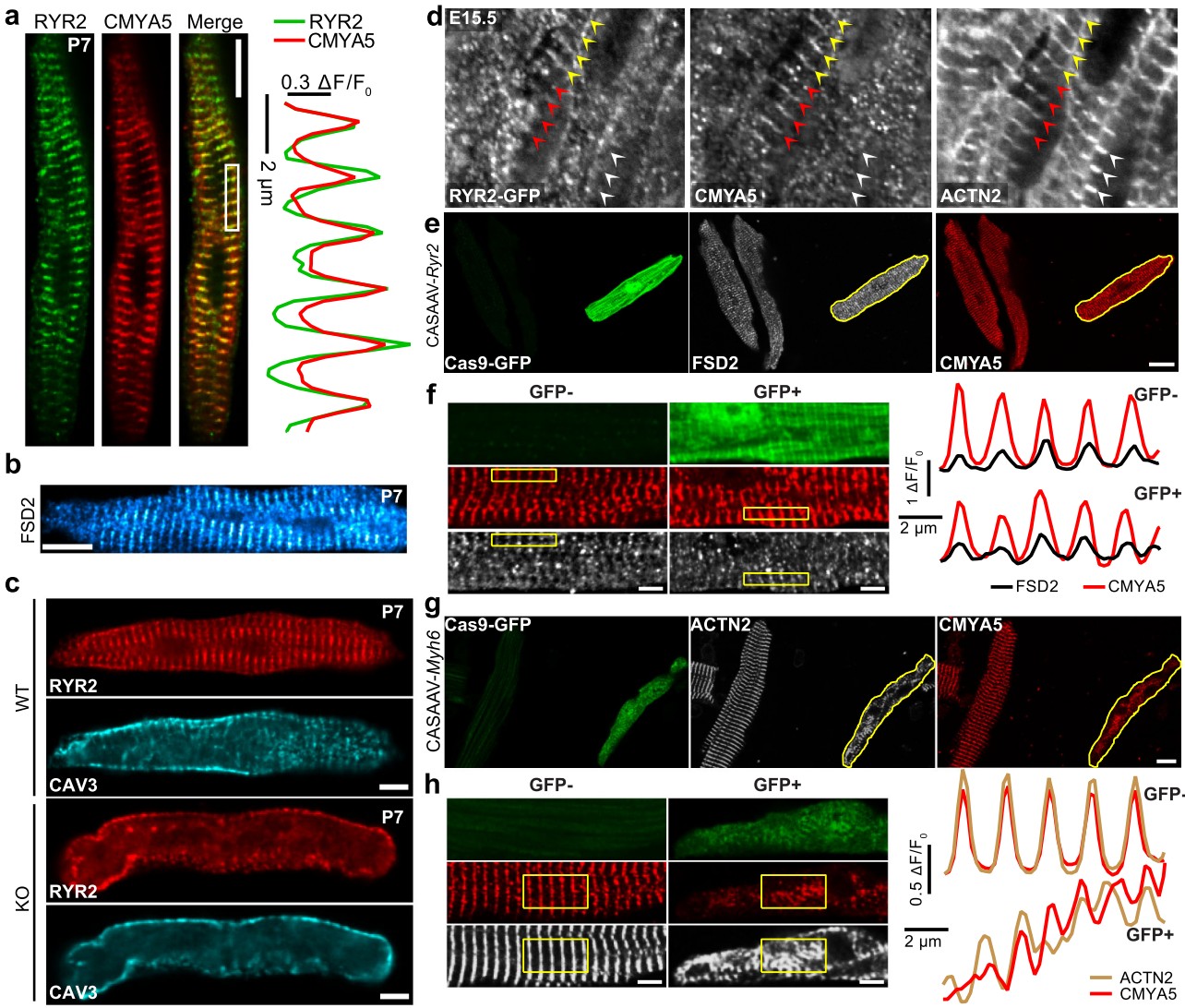

**Fig. 3 CMYA5 positions jSR adjacent to Z-lines. a** P7 ventricular cardiomyocyte co-immunostained for CMYA5 and RYR2. Bar, 10 μm. Bottom, spatial profile plot demonstrates co-localization of CMYA5 and RYR2 in a striated pattern. **b** Localization of FSD2 in P7 ventricular cardiomyocytes. FSD2 adopted a striated pattern. Bar, 10 μm. Representative of two independent experiments. **c** jSR (RYR2) and T-tubule (CAV3) organization in WT and KO P7 ventricular cardiomyocytes. CMYA5 ablation caused loss of jSR organization. At this stage, T-tubules were not yet present in either genotype. Bar, 10 μm. Representative of three independent experiments. **d** RYR2, CMYA5, and ACTN2 localization in WT E15.5 ventricular myocardium. Assembling sarcomere Z-lines (ACTN2) co-localized with CMYA5 alone (red arrowheads), CMYA5 and RYR2 (yellow arrowheads), or neither (white arrowheads). Bar, 5 μm. Representative of five independent experiments. **e, f** Effect of CASAVV-mediated ablation of RYR2 on CMYA5 and FSD2 localization. CASAAV somatic mutagenesis was used to deplete RYR2 in a subset of cardiomyocytes. CMYA5 and FSD2 localization was evaluated in RYR2-deficient (GFP+) and control (GFP−) cardiomyocytes. Spatial profiles of boxed areas in **f**, plotted at right, demonstrate that RYR2 ablation did not impact CMYA5 or FSD2 localization. Bar = 10 μm (**e**), 5 μm (**f**). Representative of three independent experiments. **g, h** Effect of CASAAV-mediated ablation of MYH6 on CMYA5 and ACTN2 localization. CASAAV was used to deplete MYH6 in a subset of cardiomyocytes. MYH6-deficient (GFP+) had impaired sarcomerogenesis and disorganization of Z-line marker ACTN2, and CMYA5 organization was correspondingly deranged. Signal intensities in boxed areas in **h** are plotted to the right. Bar = 10 μm (**g**), 5 μm (**h**). Representative of three independent experiments.

but not in *Cmya5* KO (Supplementary Fig. 7). Neither CMYA5[MD9] nor FSD2 rescued *Cmya5* KO T-tubule organization (Supplementary Fig. 7a) or heart systolic function (Supplementary Fig. 7b), suggesting that portions of CMYA5 within the N-terminal portion of the protein contribute to CMYA5 localization. With the exception of amino acids 78-319, CMYA5's N-terminus is poorly conserved[34]. When expressed in cardiomyocytes in vivo, CMYA5[1-450] and CMYA5[1-1200], which share the conserved region, co-localized well with Z-line marker ACTN2 (Supplementary Fig. 7c). These data indicate that the N-terminal portion of CMYA5 likely promotes its localization at Z-lines and its role in maintaining heart function.

**CMYA5 regulates normal dyad Ca²⁺ dynamics.** Dyads play a central role in E-C coupling, and *Cmya5* KO disrupts dyad structure and positioning. To assess the functional consequence of *Cmya5* KO on cardiomyocyte Ca²⁺ signaling, we visualized Ca²⁺ dynamics at individual dyads in contracting cardiomyocytes using a nanospark sensor localized to the dyadic cleft (GCaMP6f-junctin; abbreviated ASPH-G6f)[22]. After AAV-mediated expression of ASPH-G6f in adult heart, we measured spontaneous jSR Ca²⁺ release events in freshly isolated mature cardiomyocytes. KO cardiomyocytes had elevated Ca²⁺ spark frequency, which was strongly augmented by β-agonist (100 nM isoproterenol; ISO; Fig. 4a, b). Next, we examined junctional Ca²⁺ dynamics during

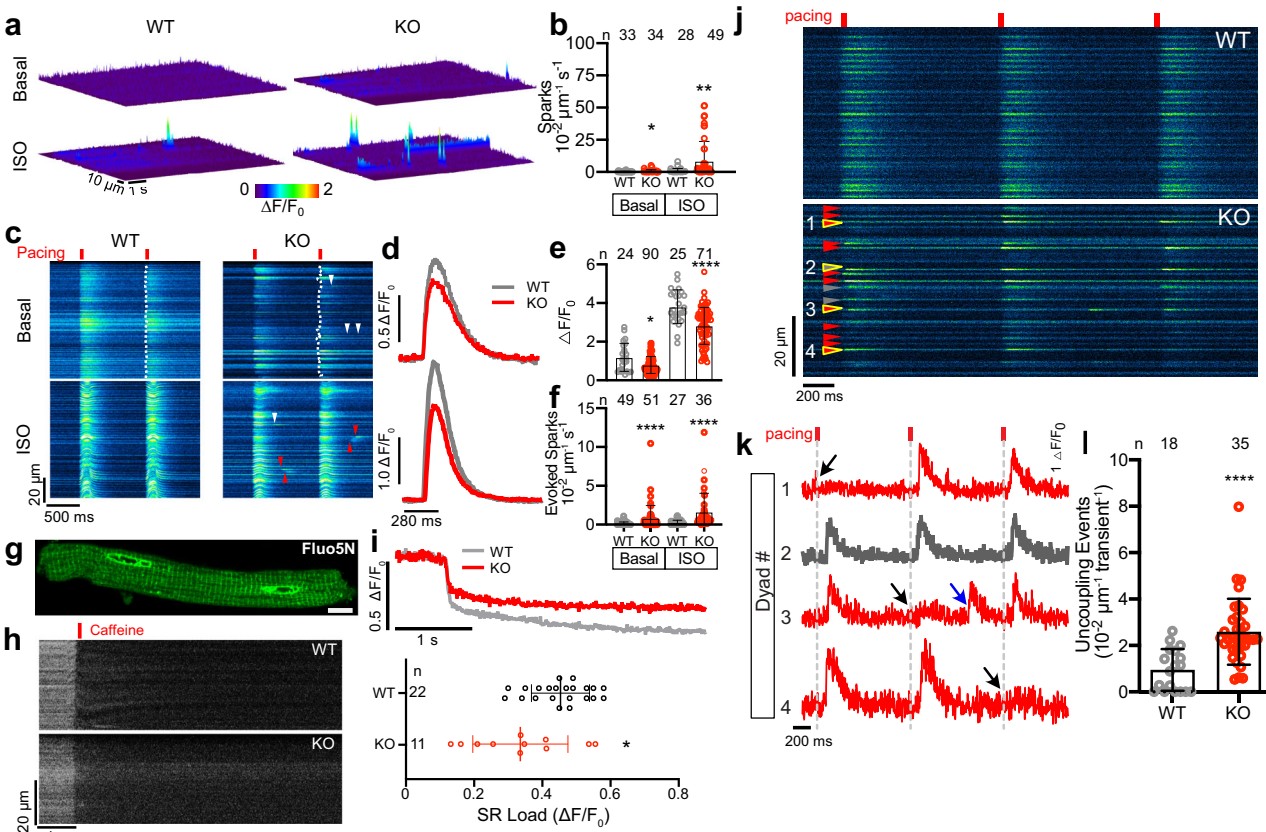

**Fig. 4 Altered dyadic Ca$^{2+}$ release in CMYA5 KO cardiomyocytes.** Ca$^{2+}$ release in mature *Cmya5* KO or WT ventricular cardiomyocytes was assessed using a dyad-localized Ca$^{2+}$ nanosensor, GCaMP6f-Junctin (ASPH-G6f) and detected by confocal line scan imaging. **a, b** Surface plots of Ca$^{2+}$ sparks in basal and isoproterenol (ISO)-stimulated (100 nM) cardiomyocytes. Quantification is shown in (b). Mann-Whitney test within basal or ISO conditions. **c**–**f** Ca$^{2+}$ release during electrical pacing (red lines) under basal conditions or ISO stimulation. Dotted line, leading edge of the evoked Ca$^{2+}$ transient. Arrowheads, aberrant Ca$^{2+}$ release outside of the initial evoked Ca$^{2+}$ transient, without (white) or with (red) wave-like propagation. d. Representative Ca$^{2+}$ transients under basal (top) or ISO (bottom) conditions. **e, f** Quantification (mean ± SD) of Ca$^{2+}$ transient amplitude and Ca$^{2+}$ spark frequency. Mann-Whitney test within basal or ISO conditions. **g**–**i** Measurement of SR Ca$^{2+}$ stores using low-affinity Ca$^{2+}$-sensitive dye Fluo5N and AAV-mediated expression of SR-targeted esterase, which trapped Fluo5N in SR. g, Fluo5N distribution. **h, i** SR Ca$^{2+}$ release induced by 10 mM caffeine. **g**–**i** are representative of three independent experiments. Caffeine-induced change in Fluo5N signal yielded an estimate of SR Ca$^{2+}$ stores (mean ± SD). *t* test. Bar = 10 μm. **j** Ca$^{2+}$ release at dyads of electrically paced cardiomyocytes was recorded using the ASPH-G6f dyad-targeted Ca$^{2+}$ nanosensor. Gray arrowheads, consistently coupled dyads (activated with each electrical pacing event). Red arrowheads, inconsistently coupled dyads (activated with some pacing events but not others). **k** Fluorescence intensity profiles over time of yellow outlined dyads in **j**, numbered 1–4. Dyad 2 was consistently coupled, whereas dyads 1, 3, and 4 were inconsistently coupled. Black arrows, lack of activation with electrical pacing. Blue arrow, evoked Ca$^{2+}$ spark not coordinated with the overall Ca$^{2+}$ transient. **l** Frequency of inconsistently coupled dyads in WT and KO cardiomyocytes, with and without ISO stimulation. The number of pacing events without dyadic Ca$^{2+}$ release over three consecutive calcium transients was normalized to line scan length. *n*, number of cardiomyocytes. Mann–Whitney test within basal or ISO conditions. *, *P* < 0.05; **, *P* < 0.01; ****, *P* < 0.0001. Two-sided statistical tests were used. Data are presented as mean ± SD.

electrically evoked E-C coupling. Representative results in Fig. 4c–l illustrate several abnormalities in KO cardiomyocytes. First, KO CMs exhibited reduced Ca$^{2+}$ transient amplitude under both normal and ISO-treated conditions (Fig. 4c–e). This reduction is largely attributable to ~25% reduced SR Ca$^{2+}$ content, measured in intact CMs by sequestering a low-affinity Ca$^{2+}$-sensitive dye in SR through AAV-mediated expression of an SR-targeted esterase[35] (Fig. 4g–i). Second, the spatiotemporal coordination of dyadic Ca$^{2+}$ transients was impaired in KO CMs. In control cardiomyocytes, dyad Ca$^{2+}$ release during full-fledged Ca$^{2+}$ transients was temporally synchronized across adjacent dyads (Fig. 4c, WT: note smooth leading edge marked by dotted line). In contrast, Ca$^{2+}$ release in KO cardiomyocytes had reduced synchronization across adjacent dyads (Fig. 4c, KO: note ragged leading edge marked by dotted line). Ca$^{2+}$ release at some dyads was delayed relative to the overall initiation of the Ca$^{2+}$

transient, making the transient less abrupt. This variability in dyad Ca$^{2+}$ release is consistent with the architectural abnormalities that we observed in KO dyads. Third, the dyadic Ca$^{2+}$ nanosensor revealed that a subset of *Cmya5* KO dyads became intermittently uncoupled, i.e. some KO dyads activated inconsistently during electrical pacing (Fig. 4j, k). Quantitative analysis confirmed that *Cmya5* KO dyads were significantly more likely to exhibit inconsistent coupling (Fig. 4l). This finding agrees with reduced Ca$^{2+}$ release synchronization (Fig. 4c) and the abnormal and more heterogeneous dyad architecture (Fig. 2j–l). Finally, during electrical pacing, we observed more frequent evoked Ca$^{2+}$ sparks interspersed between fully activated Ca$^{2+}$ transients in KO cardiomyocytes (Fig. 4c, top arrowheads; Fig. 4k, blue arrow; quantified in Fig. 4f). Under ISO stimulation, these Ca$^{2+}$ sparks became even more frequent and turned into wave-like propagations (Fig. 4c, bottom; arrowheads; Fig. 4f) that could increase the

propensity for $Ca^{2+}$-dependent arrhythmias. Collectively, these results demonstrate that CMYA5 is required to coordinate E-C coupling and regulate RYR2 activity.

**CMYA5 stabilizes dyad structure and function to biomechanical stress.** Biomechanical stress on the heart promotes T-tubule and dyad disorganization[4,9,10]. Therefore we tested the hypothesis that CMYA5 stabilizes dyads in the context of biomechanical stress by analyzing the response of *Cmya5* KO hearts to pressure overload, induced by surgical transverse aortic constriction (TAC). WT, Het, and KO mice were subjected to TAC or Sham operation, followed by weekly echocardiography. Physiological and histological studies were performed 4 weeks after TAC. After TAC, WT mice developed mild ventricular dysfunction which remained stable between weeks 1–4. In contrast, KO + TAC mice developed more severe ventricular dysfunction at 1 week, which became progressively more severe in subsequent weeks (Fig. 5a). This ventricular dysfunction was accompanied by adverse remodeling, manifested as progressive ventricular dilatation (Supplementary Fig. 8a, b). At necropsy, KO + TAC hearts appeared hypertrophied (Fig. 5b), and this was confirmed by gravimetric measurements (Fig. 5c). In Sham-operated mice, the area of fibrotic tissue was comparable between genotypes (Supplementary Fig. 8c). TAC induced fibrosis in all genotypes compared to Sham (Supplementary Fig. 8d); KO mice developed 2.8-fold more fibrosis than wild type ($P < 0.0001$), and Het mice also had a significantly greater fibrotic area ($P < 0.01$, 1.3-fold greater than WT). Consistent with increased fibrosis, we also detected increased apoptosis after TAC in both Het and KO compared to WT (Supplementary Fig. 8e, f).

To assess the effect of TAC and CMYA5 ablation on dyad architecture and positioning, we examined cardiomyocyte ultra-structure using TEM. Dyads in KO + TAC cardiomyocytes were highly abnormal (Fig. 5d). In terms of quantitative metrics, TAC induced abnormal T-tubule circularity in all genotypes, with KO and Het being more severely affected than WT (Fig. 5e). TAC significantly increased Junction-to-Z-line distance in KO compared to WT, and significantly reduced the coupling ratio to a comparable level across all genotypes (Fig. 5f, g).

We assessed dyad function by measuring dyadic $Ca^{2+}$ release in cardiomyocytes expressing the ASPH-G6f dyadic $Ca^{2+}$ nanosensor. The nanosensor adopted the expected striated localization in WT + Sham, which was less organized in WT + TAC, consistent with the known disruptive effect of TAC on dyads[9] (Fig. 5h). KO + Sham dyads were also disorganized, and this disarray was further exacerbated by TAC (Fig. 5h). $Ca^{2+}$ sparks, recorded using the ASPH-G6f nanosensor, were more frequent in KO than WT after TAC (Fig. 5i). ISO further amplified the deleterious effects of KO on $Ca^{2+}$ spark frequency. Taken together, we conclude that CMYA5 is required to organize both jSR and T-tubular components of dyads and is essential for cardiomyocyte structural and functional integrity in the face of biomechanical stress.

## Discussion

Cardiac dyads, nanodomains formed by the close apposition of T-tubules and jSR membranes, are critical to E-C coupling[1,3]. The mechanisms that establish dyad architecture and that position dyads at sarcomere Z-lines have been an unsolved mystery. Given the importance of these structures for E-C coupling, efficient cardiac contraction, and cardiac rhythm, this puzzle has critical implications for cardiac homeostasis and disease. Here we revealed that CMYA5 plays an important role in a hierarchical process of dyad assembly, in which CMYA5 localizes to sarcomere Z-lines, even in fetal cardiomyocytes that lack T-tubules or

jSR. In neonatal cardiomyocytes, jSR marked by RYR2 co-localizes with CMYA5 at Z-lines in the absence of T-tubules, which develop later. These results, and prior experiments demonstrating that CMYA5 forms a protein complex with RYR2 and influences its intracellular clustering[12], indicate that CMYA5-RYR2 interaction tethers jSR to Z-lines. When T-tubules form in juvenile cardiomyocytes, they co-localize with jSR through incompletely understood mechanisms involving JPH2[6] and RYR2[7], forming dyads. Thus, our studies provide an integrated developmental time course of dyad assembly and identify CMYA5 as a core component of the mechanisms that establish the subcellular localization of dyads at Z-lines.

Functionally, in keeping with its key role in organizing dyad architecture and localization, we demonstrate that CMYA5 is critical for regulating cardiomyocyte $Ca^{2+}$ release. CMYA5 ablation reduced the synchronization and amplitude of dyadic $Ca^{2+}$ release, which likely contributed to the impaired ventricular function of *Cmya5* KO hearts. We also observed more frequent spontaneous $Ca^{2+}$ release in *Cmya5* KO hearts, especially with β-adrenergic stimulation. These data indicate that CMYA5 regulates RYR2 activity, through direct physical interaction[12], by governing RYR2 clustering[36], or by recruiting proteins that modulate RYR2 activity, such as Protein Kinase A[18,37,38].

Despite the strong effects of CMYA5 ablation on dyad architecture and organization within cardiomyocytes, *Cmya5* KO hearts had only mildly impaired baseline contractile function. This is unlikely to be due to redundancy with the structurally related FSD2, since FSD2 overexpression did not rescue *Cmya5* KO hearts and FSD2 was unable to properly localize in the absence of CMYA5. Rather, it likely reflects the considerable contractile reserve of the mammalian heart and genetic compensation for chronic *Cmya5* ablation. Stressing the heart by aortic constriction unmasked the profound effect of CMYA5 ablation. CMYA5 KO hearts developed severe, progressive ventricular dysfunction and underwent adverse remodeling. These organ level findings were linked to greater disarray of dyads in CMYA5 KO induced by TAC. Particularly notable was the marked increase in $Ca^{2+}$ sparks, especially with β-adrenergic stimulation, which may be related to the propensity of failing hearts to develop ventricular arrhythmias. Together these data show that CMYA5 is important for the structural and functional integrity of cardiac dyads, particularly under stress. These findings suggest that CMYA5-dependent mechanisms may become dysregulated in diseased hearts, leading to loss of dyad architecture and function.

While this manuscript was in preparation, Tsoupri et al. reported the cardiac phenotype of an independently generated *Cmya5* knockout mouse line[19]. Consistent with our study, Tsoupri et al found that *Cmya5* knockout caused mild cardiac dysfunction, mislocalization of RYR2, and abnormal T-tubule and jSR morphology. Our study makes several conceptual advances beyond the work of Tsoupri et al. Our detailed studies of CMYA5 reveal its significant contribution to a hierarchical and developmentally ordered process of dyad assembly, in which CMYA5 tethers jSR to Z-lines, and with the subsequent addition of T-tubules. In addition to its critical role in establishing normal dyad architecture, CMYA5 regulated dyad function by enhancing the fidelity of E-C coupling and by limiting spontaneous RYR2 $Ca^{2+}$ release. Finally, we show that CMYA5 protects dyad structure and function from biomechanical stress. Future investigations are warranted to determine how CMYA5 coordinates with other proteins to direct the formation, maintenance, and positioning of dyads, and how CMYA5 regulates $Ca^{2+}$ release from RYR2.

## Methods

**Animal.** Experiments were conducted compliant with all relevant ethical regulations. All animal experiments were performed under protocols approved by the

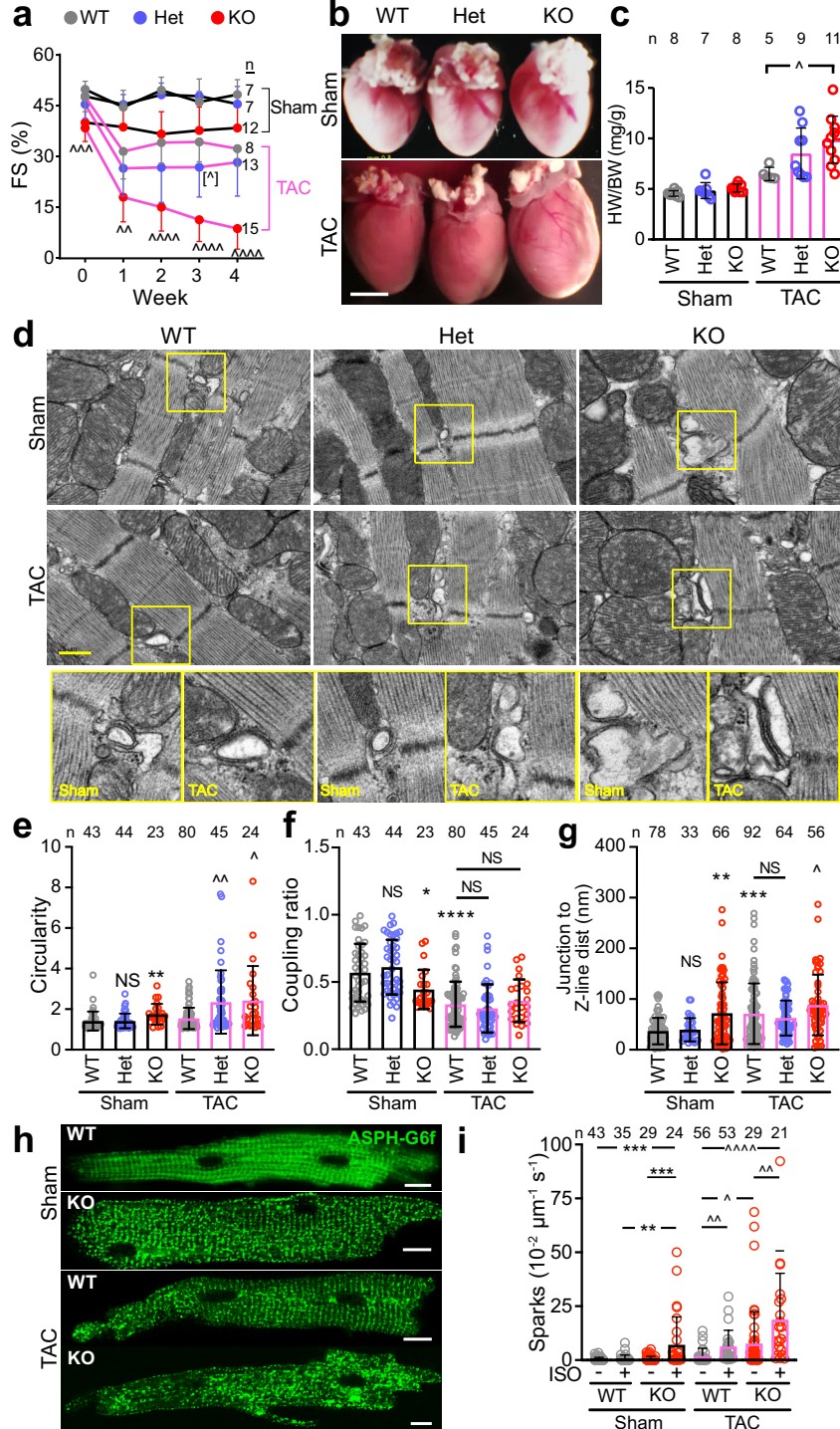

**Fig. 5 CMYA5 protects the heart and dyads from the deleterious effects of pressure overload.** *Cmya5* KO and WT mice underwent baseline echocardiography (week 0) and then TAC or Sham surgery. After 4 weekly echocardiograms, necropsy was performed. **a** Systolic heart function. FS fractional shortening. Repeated-measures two-way ANOVA was performed for TAC cohort, with a comparison to WT-TAC (^) at each time point. *n*, number of mice. **b** Gross cardiac morphology. Bar, 2.5 mm. **c** Heart weight normalized to body weight. Kruskal-Wallis with Dunn's multiple comparison test vs. WT within Sham (*) or TAC (^) cohorts. *n* number of mice. **d–g** Dyad architecture analyzed by transmission electron microscopy. Boxed regions are magnified in the bottom row. Bar, 500 nm. Quantification of T-tubule circularity, jSR-to-T-tubule coupling ratio, and Junction-Z-line distance. Kruskal–Wallis with Dunn's multiple testing correction vs. WT Sham (*) or TAC (^). *n*, number of dyads per group, from at least 10 cardiomyocytes from 3 different mice. **h**, **i** Cardiomyocytes were transduced with dyad-localized ASPH-G6f $Ca^{2+}$ nanosensor. Images of nanosensor distribution indicate dyad disorganization induced by TAC and *Cmya5* KO (**h**). Quantification of $Ca^{2+}$ sparks by confocal line scan imaging of nanosensor. Kruskal–Wallis with Dunn's multiple testing correction within Sham (*) or TAC (^). *n*, number of cardiomyocytes per group. ^, $P = 0.054$; ^ or *, $P < 0.05$; ^^ or **, $P < 0.01$; ^^^ or ***, $P < 0.001$; ^^^^ or ****, $P < 0.0001$. Two-sided statistical tests were used. Data are presented as mean ± SD.

Boston Children's Hospital Institutional Animal Care and Use Committee. *Cmya5*$^{\Delta/\Delta}$ (C57BL/6NJ-*Cmya5*$^{em1(IMPC)J}$/Mmja; Stock no. 032826) and *Rosa-Cas9GFP/Cas9GFP* (*Gt(ROSA)26Sor*$^{tm1(CAG-cas9*,-EGFP)Fezh}$; Stock No. 026175)[39] were obtained from the Jackson Laboratory. *Ryr2-GFP* knockin mice were described previously[31]. Mice were housed in a specific pathogen-free room with 12 h/12 h light/dark cycle, temperature (22 ± 2 °C), and relative humidity (45–65%). Mice were genotyped using PCR primers shown in Supplementary Table 1.

**Echocardiography.** Echocardiography was performed on a VisualSonics Vevo 2100 instrument with Vevostrain software by an investigator blinded to genotype or treatment group. Animals were awake during the procedure and held in a standard handgrip. The echocardiographer was blinded to group assignment.

**Transverse aortic constriction.** Aortic banding was performed on male mice between 25 and 30 g using a previously described protocol[40]. Mice were anesthetized with isoflurane, intubated, and mechanically ventilated. The chest cavity was entered through an incision in the left second intercostal space. The transverse aorta was dissected from the surrounding tissues. A silk suture was passed underneath the aorta and ligated against a 27 gauge needle between the brachiocephalic trunk and the left common carotid artery. The needle was then removed, resulting in a ligature with a fixed diameter constricting the aorta. The chest cavity, muscles, and skin were closed layer by layer. The sham operation was identical except that the aorta was not manipulated. The surgeon was blinded to genotype.

**Virus production and adeno-associated virus (AAV) injection.** For overexpression experiments, genes of interest were subcloned into AAV9-cardiac troponin T promoter vector[41] (Addgene, #69915). myc-BirA*[20] was from Addgene (#35700); R-CEPIA1er[42] was from Addgene (#58216); FSD2 was from Horizon Discovery, (#OMM5895-202524631); CMYA5$^{MD9}$ was synthesized by Genewiz. For CASAAV[7], gRNAs-containing oligonucleotides (Supplementary Table 1) were cloned into AAV-U6-gRNA-Tnnt2-Cre vector[7] (Addgene, #132551). For AAV production, the AAV plasmids along with the helper plasmids AAV9-Rep/Cap (Addgene, #112865) and pAd-ΔF6 (Addgene, #112867) were co-transfected into HEK293T cells using polyethylenimine (Polysciences, 23966-2). AAV was purified by iodixanol density gradient centrifugation[43]. The density gradient was made by layering the following solutions in Optiseal tubes (BECKMAN COULTER, 362183): 6 mL of 17% iodixanol, 5 mL of 25% iodixanol, 4 mL of 40% iodixanol, 5 mL of 60% iodixanol. HEK293T cell lysate containing AAV was layered on top of the density gradient and centrifuged at 45000 g for 2 h. AAV was recovered from the density gradient and titered by qPCR. AAV was then injected into P1 (postnatal day 1) pups subcutaneously in a total volume of <30 μL at a mosaic dose of $1 \times 10^9$ viral genomes per gram body weight (vg/g), at a middle dose of $2 \times 10^{10}$ vg/g, or at a full dose of $5.5 \times 10^{10}$ vg/g.

**BioID proximity proteomics.** In all, 0.5 mg protein extracts were immunoprecipitated using streptavidin Dynabeads (Invitrogen, #M280). The beads were washed 5 times with RIPA lysis buffer (Santa Cruz,# sc-24948) and stored in PBS for on-bead digestion. Liquid chromatography with tandem mass spectrometry was performed at the Taplin Biological Mass Spectrometry Facility, Harvard Medical School. Three hearts were used for streptavidin pull-down in each group. Beads were washed at least five times with 100 μL 50 mM ammonium bicarbonate. Then 5 μL (200 ng/μL) of modified sequencing-grade trypsin (Promega, Madison, WI) was spiked in, and the samples were incubated at 37 °C overnight. The beads were then removed using a magnet, and the supernatant was dried in a speed-vac. The samples were re-suspended in 50 μL HPLC solvent A (2.5% acetonitrile, 0.1% formic acid) and desalted by STAGE tip[44]. On the day of analysis, the samples were reconstituted in 10 μL of HPLC solvent A. A nano-scale reverse-phase HPLC capillary column was created by packing 2.6 μm C18 spherical silica beads into a fused silica capillary (100 μm inner diameter × ~30 cm length) with a flame-drawn tip[45]. After equilibrating the column, each sample was loaded via a Famos autosampler (LC Packings, San Francisco CA). A gradient was formed, and peptides were eluted with increasing concentrations of solvent B (97.5% acetonitrile, 0.1% formic acid). As peptides eluted, they were subjected to electrospray ionization and then entered an LTQ Orbitrap Velos Elite ion-trap mass spectrometer (Thermo Fisher Scientific, Waltham, MA). Peptides were detected, isolated, and fragmented to produce a tandem mass spectrum of specific fragment ions for each peptide. Peptide sequences (and hence protein identity) were determined by matching protein databases with the acquired fragmentation pattern by the software program, Sequest (Thermo Fisher Scientific, Waltham, MA)[46]. All databases include a reversed version of all the sequences and the data was filtered to between a one and two percent peptide false discovery rate.

**RNA extraction and quantitation.** Total RNA was isolated using Direct-zol RNA Miniprep Plus Kits (Zymo Research, #R2071). 1 μg of DNase I-pretreated RNA was used as input for reverse transcription using SuperScript$^{TM}$ III First-strand Synthesis SuperMix (Thermo Fisher Scientific, #18080400). Gene expression was analyzed by qPCR using Power SYBR Green PCR Master Mix (Applied Biosystems, #4367659) and a Bio-Rad CFX96 touch thermocycler, using qPCR primers listed in Supplementary Table 1.

**Western blotting.** In all, 50 μg cell lysates or nuclear extracts[47] were separated by 4–12% SDS-PAGE (Invitrogen, #NW04120BOX) and transferred to Immobilon-P PVDF membranes (Merck Millipore, #IPVH00010). The membranes were blocked with 5% nonfat dry milk and incubated with primary antibody (Supplementary Table 2) overnight at 4 °C, followed by incubation with secondary antibody for 1 h at room temperature. Western blot signals were captured using a Fujifilm LAS-3000 imager and quantified using Fiji.

**In situ confocal imaging of cardiomyocyte T-tubule structure in intact hearts.** For in situ T-tubule imaging[27], intact mouse hearts underwent Langendorff perfusion at room temperature with 0 Ca$^{2+}$ Tyrode solution (in mM, pH 7.4: NaCl 137, glucose 15, HEPES 20, KCl 4.9, MgCl$_2$ 1.2, NaH$_2$PO$_4$ 1.2), containing 2.5 μM MM 4-64 (Enzo Life Sciences, # ENZ-52252), a lipophilic fluorescence indicator of membrane structure, for 20 min. The hearts were then placed in a perfusion chamber mounted on the stage of an Olympus FV3000RS confocal microscope and imaged in situ with ×60 (NA = 1.4) oil immersion lens. The optical pinhole was set to 1 airy disc (<1 μm axial resolution). Excitation for MM4-64 was 488 nm, and emission was 680–780 nm.

**Cardiomyocyte isolation.** Single ventricular or atrial myocytes were enzymatically isolated from mouse hearts. Hearts were excised from isoflurane-anesthetized animals, rinsed in cold perfusion buffer (PB, in mM, pH 7.4: NaCl 137, glucose 15, HEPES 20, KCl 4.9, MgCl$_2$ 1.2, NaH$_2$PO$_4$ 1.2, taurine 5, and 2,3-Butanedione monoxime (BDM) 10), and quickly mounted on a Langendorff perfusion system. Hearts were then perfused at 37 °C with oxygenated Ca$^{2+}$-free PB (gassed with 95% O$_2$, 5% CO$_2$) until blood was completely cleared (about 5 min). Solution was then switched to the digestion solution (PB containing 0.7 mg/mL Type II Collagenase (Worthington, #LS004176), 0.1 mg/mL Type XIV Protease (Sigma-Aldrich, #P5147), 1 mg/mL bovine serum albumin (BSA, Sigma-Aldrich, #A3912) and 50 μM Ca$^{2+}$). Once hearts became soft (about 15–30 min), perfusion was stopped and ventricles or atria were gently minced into small pieces and agitated by blunt-tipped transfer pipettes in PB containing 1 mg/mL BSA and 50 μM Ca$^{2+}$. Cells were filtered by 100 μm cell strainer (Fisher Scientific, #0877119), centrifuged for 1 min at 20 g and subjected to Ca$^{2+}$ gradient recovery using PB containing 1 mg/mL BSA and Ca$^{2+}$ from 0.25, 0.35, 0.52 to 1 mM sequentially.

Coverslips were pre-coated with 5 μg/mL laminin (Sigma-Aldrich, #L2020) for at least 30 min at 37 °C. Freshly isolated myocytes were plated onto the coverslips for 30 min in the incubator. Attached mouse CMs were cultured in MEM medium (Sigma-Aldrich, #M2279) supplemented with 10 mM BDM.

**Ca$^{2+}$ dye loading and live-cell imaging.** To simultaneously record cytosolic and SR Ca$^{2+}$ signals[35], freshly isolated CMs from AAV9-Tnnt2-srCES2 injected mice were incubated with low-affinity Ca$^{2+}$ indicator Fluo5N-AM (5 μmol/L, Invitrogen, #F14204) at 37 °C for 10 min, washed once, and incubated with Rhod2-AM (5 μmol/L, Invitrogen, #R1244) at 37 °C for 8 min, and then gently washed two times. Dye loading, washing, and Ca$^{2+}$ imaging were conducted in Tyrode solution containing 1 mM Ca$^{2+}$.

Confocal imaging was performed with an Olympus FV3000RS microscope with a ×60 1.4 NA oil immersion objective and a line scan speed of 3.78 ms/line. The pinhole was set for a nominal 1 μm optical section. For single-channel measurement of ASPH-G6f, excitation was at 488 nm, and fluorescence emissions were collected at between 490 and 540 nm. For simultaneous measurement of Fluo5N and Rhod2, excitation was at 488 and 543 nm, and fluorescence emissions were collected at between 490 and 520 nm and >560 nm, respectively. For Ca$^{2+}$ transients recording, CMs were perfused with Tyrode solution containing 10 mM butanedione monoxime to avoid motion artifacts, and field stimulation was applied at 1 Hz.

**Histologic and immunofluorescent assays.** Hearts were fixed in 4% paraformaldehyde (PFA) overnight and allowed to sink in 30% sucrose (typically 3–4 h) prior to freezing in tissue freezing medium (TFM, General Data). 10 μm thick cryosections were affixed to slides. Heart sections were permeabilized with 0.5% Triton X-100 for 20 min and blocked with 10% normal donkey serum for 1 h. Seeded isolated cells were fixed in 2% PFA for 15 min at room temperature, washed 3 times with PBS, permeabilized with 0.1% Triton X-100 for 10 min, then rinsed 3 times with PBS, and blocked with 1% BSA for 1 h. The sections or cells were then incubated with primary antibody (Supplementary Table 2) overnight at 4 °C, washed with PBS 3 times, incubated with Alexa fluor dyes-conjugated donkey secondary antibody (Invitrogen, 1:200) for 2 h at room temperature, washed with PBS 3 times, and then incubated with 1 mg/mL 4', 6-diamidino-2-phenylindole (DAPI) for 10 min at room temperature. Immunofluorescence staining was visualized using a confocal microscope at 405 nm (DAPI), 488 nm (Alexa fluor 488), 543 nm (Alexa fluor 555) and 647 nm (Alexa fluor 647) excitation, and 420–470 nm, 490–520 nm, 560–620 nm and > 650 nm emission, respectively.

For proximity ligation assay (PLA), PFA (4%)-fixed CMs were permeabilized with PBS containing 0.1% Triton X-100. Duolink PLA was carried out using the

Duolink® In Situ Red Starter Kit Mouse/Rabbit (Sigma-Aldrich, #DUO92101-1KT).

For fibrosis and apoptosis measurement, mouse hearts were excised and fixed in 4% PFA overnight, dehydrated through ethanol, embedded in paraffin, and sectioned at 4 μm. Sections were dewaxed, rehydrated, post-fixed, and subject to Masson trichrome staining for fibrosis visualization and TUNEL staining (Invitrogen, #C10617) for in situ apoptosis detection. Sections were imaged by a widefield microscope (Keyence) at ×10 magnification, or a laser scanning confocal (Olympus FV3000RS). Fiji was used for the quantification.

**Electron microscopy (EM)**. EM experiments were performed by the Electron Microscopy Core in Beth Israel Deaconess Medical Center, Harvard Medical School. Briefly, heart samples were collected, cut into small pieces (1–2 mm cubes), and fixed in EM fixative (2.5% Glutaraldehyde, 2.5% PFA in 0.1 mol/L sodium cacodylate buffer, pH 7.4) overnight at 4 °C. After fixation, tissues were washed and post-fixed, dehydrated, and infiltrated with resin. After curing the resin, thick sections (0.5 μm) were cut and stained with Toluidine Blue prior to thin sectioning. Ultrathin sections (70–90 nm) were placed onto copper grids stained with uranyl acetate and lead citrate and examined by a JEOL1400 transmission electron microscope.

**RNA-seq**. Single-cell RNA-seq data on normal human hearts were obtained from GSE109816 using the provided unique molecular identifier matrix[25].

**Image processing and analysis**. T-tubule[9] and $Ca^{2+}$ imaging[35] data were analyzed using custom code written in Interactive Data Language (ITT, New York, NY). The code is provided in the Supplementary Materials. EM images were analyzed to determine T-tubule circularity, coupling ratio, and Junction-Z-line distance, as described in Supplementary Fig. 4a and ref. [10].

**Statistics and reproducibility**. Each experiment was repeated independently with similar results at least two times, or as specified in the figure legends. Measurements were made blinded to group assignment. Two-sided Student's $t$ test and analysis of variance were used for normally distributed data, and two-sided Mann–Whitney or Kruskal–Wallis non-parametric tests were used otherwise. $P < 0.05$ was considered statistically significant. Statistical analysis was performed using Graphpad Prism 9. Results are displayed as mean ± standard deviation. Sample sizes indicate independent biological replicates.

**Reporting summary**. Further information on research design is available in the Nature Research Reporting Summary linked to this article.

## Data availability
Data for this manuscript are provided in the figures, Supplementary Figures, Tables, Data, and code. The mass spectrometry proteomics data generated in this study have been deposited in the PRIDE database under accession code PXD028960. The sequencing data used in this study are available in the GEO database under accession code GSE109816. Source data are provided with this paper.

## Code availability
Custom code used for image processing is provided in the Supplementary Materials.

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

## Acknowledgements

We thank Ross Tomaino from the Taplin Biological Mass Spectrometry Facility for the mass spectrometry analysis. F.L. was supported by American Heart Association Postdoctoral Fellowship 20POST35200233. W.T.P. was supported by NIH grants R01 HL146634 and R21 HD094909.

## Author contributions

F.L. designed experiments, performed experiments, analyzed data, and wrote the manuscript. Q.M. performed surgeries and echocardiograms. W.X. contributed to data analysis. D.Z., M.E.S., B.D.J., Y.G., and F.N., and H.C. contributed ideas, samples, and reagents. C.L.L. assisted in data collection. W.T.P. oversaw the overall project, analyzed data, and edited the manuscript.

## Competing interests

The authors declare no competing interests.
