## [Peer Review File · Nature Communications]

REVIEWER COMMENTS

Reviewer #1 (Remarks to the Author):

I think the MS is interesting and although the English is awkward in places it is easy to read and suitable for increasing knowledge in general cardiac physiology. Perhaps a bit more modesty for the apparently strong claims of the role of the protein might make the MS more accurate without detracting from the central message?

Considering the title, the introduction tells the reader very little about the background knowledge of what CMYA5 is and what it is known to do. Quite a bit of work has already been done of this protein including demonstration of effects on jSR structure and cardiac function, so it is disingenuous to say (184) "little studied" in the context of the present paper. The demonstration of cardiac mitochondrial defects is also notable. (Perhaps even more intriguing are the demonstrations of both schizophrenia association and brain defects related to this gene product - doesn't this imply a function beyond tt-SR assembly?). I would also point out that ASPH has also been designated as TRIM76.

The gene symbol CMYA5, arose from involvement of myospryn in cardiomyopathy and was originally hypothetical but a link to cardiomyopathy was suggested by co-expression of myospryn with known cardiomyopathy genes (Walker 2001). This hypothesis gained empirical support from the association of a myospryn cSNP haplotype with left ventricular wall thickness and diastolic dysfunction in patients (Nakagami et al. 2007). Such background should be included.

This protein has been implicated as an anchoring protein that may serve to localize A-kinase (among other possible binding functions) so this raises the question as to whether the (moderate) changes seen in the KO model are related to this function. However, there are no controls for this possible effect/role. In this context, if beta signalling or calcineurin is affected is compromised, is it really any surprise that the failing heart is impacted and how does this support the central hypothesis?

What about interactions with titin (Hackman et al. 2002)?

The authors state "We found that CMYA5 is required to position jSR adjacent to Z-lines, an early and essential step in dyad assembly" which would imply that in the KO no dyads should form but in fact they do (see Fig. 5D).

The proteomic approach is powerful, but it is quite unclear why the authors focus on CMY5 when many other proteins that affect Ca signalling are associated with the gene product. I am struck by the fact that the highest ranked protein was phospholamban... This raises serious concerns as to causation in these experiments as many higher ranked proteins are known to affect many aspects of EC coupling.

398: Who do the authors say “GCaMP6f-junctin16 and srCES29 were kind gifts from Dr. Heping Cheng, Peking University” when Cheng is a co-author?

388: “Aortic banding was performed on male mice between 25 and 30 g, using a modification of our previously described protocol³⁵” The citation does not describe their protocol but is a general review of techniques by other authors.

Fig 1c. “Expression of BirA* dyadic biosensors in myocardium. Heart sections were stained for the myc epitope tag. The majority of cardiomyocytes were immunoreactive”. Without quantification this is an overstatement of the result which shows ~50% stained.

Fig 1E: Where are the MW markers? It is clear that the biotinylation hits many proteins and there are clear differences between triadin and junctin. Why is this ignored? I would like to have seen an overlay of the ASPH expression or at least have the putative band corresponding to the MW of ASPH indicated.

Reviewer #2 (Remarks to the Author):

These authors use an AAV-based targeted proteome biosensor to identify by mass spectrometry new proteins associated to dyads in the P1 mouse heart. CMYA5 was a protein highly enriched in virus-transduced cardiomyocytes. CMYA5 co-localizes with RYR2 in cardiomyocytes. Using PLA, the authors determine that RYR2 and CMYA5 are in close proximity, showing a striated pattern throughout cardiomyocytes. These data suggest that CMYA5 is a dyadic protein associated with RYR2. Generation of null mice by Crispr-Cas9 gene editing reveals that mutants show reduced systolic function and ventricular dilatation, indicating that CMYA5 is essential for heart function.

Dyads are critical for E-C coupling and examination of T tubules system shows that they are disrupted. T tubule disruption is a cell autonomous phenotype occurring in the absence of cardiac

dysfunction, as AAV-mediated mosaic ablation of CMYA5 in a low number of cardiomyocytes indicates. TEM reveals changes in T tubules shape and dyad architecture. The authors then study the interrelationships between T-tubules, jSR, and Z-lines in wild type and Cmya5KO adult atria and P7 and foetal ventricular cardiomyocytes. Their results show that dyads are built on scaffolding provided by sarcomeres so that CMYA5 localizes to Z-lines, and subsequently tethers jSR adjacent to these structures. T-tubules subsequently form and co-localize with jSR, yielding organized, properly positioned dyads. Then, they assay CMYA5MD9 (C-terminal region) and FSD2, using AAV vectors, for their ability to rescue Cmya5KO hearts, without success. The authors claim that the N-terminal region of CMYA5 (aa 1-2730) is essential for its localization at Z-lines.

Visualization of Ca²⁺ dynamics in contracting wild type and Cmya5KO cardiomyocytes indicated that CMYA5 is required to coordinate E-C coupling and regulate RYR2 activity. Finally, these authors use a pressure overload (TAC) model to establish that CMYA5 stabilizes dyad structure and function to biomechanical stress. Overall, the paper is very complete and has very carefully performed and elegant experiments.

One suggestion to complement the negative results with CMYA5MD9 C-terminal region is to test if, as predicted, the N-terminal region of CMYA5 (aa 1-2730) is essential for its localization at Z-lines.

Reviewer #3 (Remarks to the Author):

Review on Lu et al. "CMYA5 establishes cardiac dyad architecture and positioning"

The authors applied an innovative proximity proteomics approach in intact, living hearts to identify proteins enriched near dyads. Proteins being biotinylated in close proximity to the known transmembrane jSR proteins junctin and triadin 1 were enriched by streptavidin and identified by mass spectrometry in comparison to a control setup. Results are visualized in Figure 1C and compiled in Supplemental data 1. Data were uploaded to PRIDE database and thus will be publicly available.

The analysis looks sound. However, the authors do not comment at all the proteins identified beside CMYA5. Phospholamban (rank 1) or Striated muscle-specific serine/threonine-protein kinase (rank 2) seem to be much more interesting candidates, because CMYA5 was found only at rank 448. If the authors have used proximity proteomics only to confirm that the already –based on literature data– selected candidate CMYA5 is part of the dyad, the results part should be rewritten in this way. In the current version of the first section of the results, the selection of the candidate on which the experiments of the whole manuscript concentrates, seems to be artificial. At least the authors

should explain, why no other candidates have been taken into consideration, but that other proteins might be important in maintenance of the dyad architecture and cell integrity.

No information is provided on protein sample preparation and the number of replicates being analysed.

No information at all is provided on the LC-MS/MS method used and therefore sensitivity of the analysis cannot be evaluated.

I. 458 "oC" is missing

I. 546 "mass spectrometry" should be used instead of "mass spectroscopy"

Point-to-point Responses

Reviewer #1 (Remarks to the Author):

I think the MS is interesting and although the English is awkward in places it is easy to read and suitable for increasing knowledge in general cardiac physiology. Perhaps a bit more modesty for the apparently strong claims of the role of the protein might make the MS more accurate without detracting from the central message?

A: We thank you for the overall positive comments.

In the revised manuscript, we moderated or conditioned several of the statements as recommended by the reviewer.

Considering the title, the introduction tells the reader very little about the background knowledge of what CMYA5 is and what it is known to do. Quite a bit of work has already been done of this protein including demonstration of effects on jSR structure and cardiac function, so it is disingenuous to say (l84) "little studied" in the context of the present paper.

We added a paragraph to the introduction to briefly review what is known about this protein. The protein has been the subject of several studies and multiple interacting proteins have been reported. However, its function in muscle cells, particularly cardiomyocytes, has not been well studied, for example using genetic knockout. While we were preparing the initial submission, the first in vivo functional study was reported (Tsoupri et al, 2021). This study characterized the CMYA5 knockout mouse and reported cardiac dysfunction and abnormal ultrastructural features. One EM image showed abnormal T-tubules and jSR. However, this manuscript did not address the effect on dyad development or function, nor did it assess the dramatic effect of pressure overload on the cardiac phenotype. We included the paragraph below in the revised introduction:

CMYA5 (cardiomyopathy-associated protein5), also known as myospryn, is an under-studied ~450 kDa member of the tripartite motif-containing super-family (TRIM) that is selectively expressed in cardiac and skeletal muscle.^{11,12} TRIM proteins contain four protein-protein binding domains (RING, BBox1, BBox2, and coiled-coiled) in a conserved order and generally function as part of large protein complexes.¹³ Based on co-expression of CMYA5 with known cardiomyopathy genes, it was initially hypothetically linked to cardiomyopathy.¹⁴ This link gained empirical support from the association of a CMYA5 coding single nucleotide polymorphism with left ventricular wall thickness and diastolic dysfunction.¹⁵ CMYA5 was previously reported to interact with multiple muscle proteins, including RYR2,¹² the Z-line protein ACTN2,¹¹ desmin,¹⁶ titin,¹⁷ and protein kinase A (PKA).¹⁸ However, little has been reported about the in vivo function of CMYA5. A recent study published while this manuscript was in preparation demonstrated that CMYA5 knockout causes cardiac dysfunction and mis-localization of RYR2.¹⁹ However the effect of CMYA5 knockout on dyad formation, structure, and function was not investigated in detail.

The demonstration of cardiac mitochondrial defects is also notable. (Perhaps even more

intriguing are the demonstrations of both schizophrenia association and brain defects related to this gene product - doesn't this imply a function beyond tt-SR assembly?). I would also point out that ASPH has also been designated as TRIM76.

We checked mitochondrial alignment by in situ heart imaging with Mito-tracker Red but did not find any difference between WT and KO hearts. Further we examined the morphology of mitochondria by EM and found the mitochondria were normal. We are uncertain why our results diverge from those of Tsoupri et al. with respect to cardiac mitochondria. These data are included in the revised manuscript, Supplementary Fig. 5.

The gene symbol *CMYA5*, arose from involvement of myospryn in cardiomyopathy and was originally hypothetical but a link to cardiomyopathy was suggested by co-expression of myospryn with known cardiomyopathy genes (Walker 2001). This hypothesis gained empirical support from the association of a myospryn cSNP haplotype with left ventricular wall thickness and diastolic dysfunction in patients (Nakagami et al. 2007). Such background should be included.

We revised the introduction to include these points. Please see paragraph quoted from Introduction above.

This protein has been implicated as an anchoring protein that may serve to localize A-kinase (among other possible binding functions) so this raises the question as to whether the (moderate) changes seen in the KO model are related to this function. However, there are no controls for this possible effect/role. In this context, if beta signalling or calcineurin is affected is compromised, is it really any surprise that the failing heart is impacted and how does this support the central hypothesis?

We checked PKA localization did not find any mislocation in isolated *Cmya5* KO cardiomyocytes. We also looked at potential activation of NFAT3 by calcineurin, by measuring the extent of its nuclear localization. We did not detect a significant difference in NFAT3 nuclear localization in *CMYA5* KO heart. These data are in revised Supplementary Fig. 5.

What about interactions with titin (Hackman et al. 2002)?

CMYA5 was previously reported to interact with the C-terminus of Titin, which is located at the M-line of sarcomeres. However, in cardiomyocytes we found that *CMYA5* is localized near the Z-line, not the M-line. Furthermore, we validated that titin localization was normal in *Cmya5* KO by immunostaining in isolated CMs. These data are included in revised Supplementary Fig. 5.

The authors state "We found that *CMYA5* is required to position jSR adjacent to Z-lines, an early and essential step in dyad assembly" which would imply that in the KO no dyads should form but in fact they do (see Fig. 5D).

Although dyads are present, they are highly distorted and much of the jSR loses its

positioning adjacent to Z-lines. We stated this more precisely in the revised text: “We found that CMYA5 is required to efficiently position jSR adjacent to Z-lines...”. In other sections we state that CMYA5 regulates normal dyad assembly.

The proteomic approach is powerful, but it is quite unclear why the authors focus on CMY5 when many other protein that affect Ca signalling are associated with the gene product. I am struck by the fact that the highest ranked protein was phospholambam... This raises serious concerns as to causation in these experiments as many higher ranked protein are known to affect many aspect of EC coupling.

We prioritized mass spect hits that had no detected signal in controls. This was not well reflected in the original figure, and we revised it accordingly (revised Fig. 1f). We further prioritized hits that were consistent between Junctin and Triadin and for which relatively less was known about function in EC coupling.

Revised text:

Biotinylated proteins were purified using immobilized streptavidin and analyzed by mass spectrometry (Fig. 1f). We overlapped the genes present in both BirA*-ASPH and BirA*-TRDN groups and excluded those with similar signals in the AAV9-GFP control group. Among the proteins highly enriched in the BioID groups were RYR2, JPH2, ASPH, and TRDN (Fig. 1f; Suppl. Data 1). Recovery of these known dyadic proteins validated our experimental strategy.

We prioritized proteins found in both BioID groups that lacked signal in the control group. Among these proteins, one of the most highly enriched in the BioID groups was CMYA5, a protein expressed in striated muscle and neurons.^{23,24}

398: Who the authors say “GCaMP6f-junctin16and srCES29 were kind gifts from Dr. Heping Cheng, Peking University” when Cheng is a co-author?

We deleted this statement.

388: “Aortic banding was performed on male mice between 25 and 30 g, using a modification of our previously described protocol³⁵” The citation does not describe their protocol but is a general review of techniques by other authors.

We corrected this reference.

Fig 1c. “Expression of BirA* dyadic biosensors in myocardium. Heart sections were stained for the myc epitope tag. The majority of cardiomyocytes were immunoreactive”. Without quantification this is an overstatement of the result which show ~50% stained.

We quantified the staining and now include more representative images (Fig. 1c).

Fig 1E: Where are the MW markers? It is clear that the biotinylation hits many proteins and

there are clear differences between triadin and junctin. Why is this ignored? I would like to have seen an overlay of the ASPH expression or at least have the putative band corresponding to the mw of ASPH indicated.

In Fig. 1E, we now include molecular weight markers and label the bands that have the appropriate molecular weight (Fig. 1e). Triadin and Junctin are distinct proteins and would be expected to have different but overlapping interactomes. Experimental factors (incomplete sensitivity, false positives) can also lead to imperfect overlap between mass spect results for each bait. We prioritized interacting proteins that were consistent between Triadin- and Junctin baits.

Reviewer #2 (Remarks to the Author):

These authors use an AAV-based targeted proteome biosensor to identify by mass spectrometry new proteins associated to dyads in the P1 mouse heart. CMYA5 was a protein highly enriched in virus-transduced cardiomyocytes. CMYA5 co-localizes with RYR2 in cardiomyocytes. Using PLA, the authors determine that RYR2 and CMYA5 are in close proximity, showing a striated pattern throughout cardiomyocytes. These data suggest that CMYA5 is a dyadic protein associated with RYR2. Generation of null mice by Crispr-Cas9 gene edition reveals that mutants show reduced systolic function and ventricular dilatation, indicating that CMYA5 is essential for heart function.

Dyads are critical for E-C coupling and examination of T tubules system shows that they are disrupted. T tubule disruption is a cell autonomous phenotype occurring in the absence of cardiac dysfunction, as AAV-mediated mosaic ablation of CMYA5 in a low number of cardiomyocytes indicates. TEM reveals changes in T tubules shape and dyad architecture. The authors then study the interrelationships between T-tubules, jSR, and Z-lines in wild type and Cmya5KO adult atria and P7 and foetal ventricular cardiomyocytes. Their results show that dyads are built on scaffolding provided by sarcomeres so that CMYA5 localizes to Z-lines, and subsequently tethers jSR adjacent to these structures. T-tubules subsequently form and co-localize with jSR, yielding organized, properly positioned dyads. Then, they assay CMYA5MD9 (C-terminal region) and FSD2, using AAV vectors, for their ability to rescue Cmya5KO hearts, without success. The authors claim that the N-terminal region of CMYA5 (aa 1-2730) is essential for its localization at Z-lines.

Visualization of Ca²⁺ dynamics in contracting wild type and Cmya5KO cardiomyocytes indicated that CMYA5 is required to coordinate E-C coupling and regulate RYR2 activity. Finally, these authors use a pressure overload (TAC) model to establish that CMYA5 stabilizes dyad structure and function to biomechanical stress. Overall, the paper is very complete and has very carefully performed and elegant experiments.

We thank the reviewer for the positive and encouraging comments.

One suggestion to complement the negative results with CMYA5MD9 C-terminal region is to test if, as predicted, the N-terminal region of CMYA5 (aa 1-2730) is essential for its

localization at Z-lines.

Since AAV has a limited capacity, we were not able to test the entire 2730 amino acids that are N-terminal to CMYA5-MD9. Most of the N-terminal 2500 amino acids are predicted to be unstructured and are poorly conserved between species. However, the very N-terminal portion of CMYA5 (~aa 70-320) is conserved. However, we tested the N-terminal regions [aa1-450] and [aa1-1200], both encompassing the conserved region, for localization at Z-lines by using AAV to express them fused to HA epitope tag. As shown in revised Suppl. Fig. 6c, we found that both proteins overlapped well with Z line marker ACTN2, suggesting that the N-terminus contributes to CMYA5 localization to Z lines.

Reviewer #3 (Remarks to the Author):

Review on Lu et al. "CMYA5 establishes cardiac dyad architecture and positioning"

The authors applied an innovative proximity proteomics approach in intact, living hearts to identify proteins enriched near dyads. Proteins being biotinylated in close proximity to the known transmembrane jSR proteins junctin and triadin 1 were enriched by streptavidin and identified by mass spectrometry in comparison to a control setup. Results are visualized in Figure 1C and compiled in Supplemental data 1. Data were uploaded to PRIDE database and thus will be publicly available.

The analysis looks sound. However, the authors do not comment at all on the proteins identified beside CMYA5. Phospholamban (rank 1) or Striated muscle-specific serine/threonine-protein kinase (rank 2) seem to be much more interesting candidates, because CMYA5 was found only at rank 448. If the authors have used proximity proteomics only to confirm that the already –based on literature data- selected candidate CMYA5 is part of the dyad, the results part should be rewritten in this way. In the current version of the first section of the results, the selection of the candidate on which the experiments of the whole manuscript concentrates, seems to be artificial. At least the authors should explain, why no other candidates have been taken into consideration, but that other proteins might be important in maintenance of the dyad architecture and cell integrity.

Our proximity proteomic study was designed to identify proteins localized at or near dyads. We prioritized proteins with no signal in control samples, that were identified by both baits, and that were relatively understudied in excitation-contraction coupling. The original figure summarizing the proteomics data did not explain the important place that lack of signal in control played in prioritization of candidates. This is better illustrated in the revised figure, in which candidates are ranked by the ratio of bait signal to control signal. Using this metric, CMYA5 ranks 16th.

No information is provided on protein sample preparation and the number of replicates being analysed. No information at all is provided on the LC-MS/MS method used and therefore sensitivity of the analysis cannot be evaluated.

We now added information on the LC-MS/MS method to the Methods Section:

0.5 mg protein extracts were immunoprecipitated using streptavidin Dynabeads (Invitrogen, #M280). The beads were then washed 5 times with RIPA lysis buffer (Santa Cruz, # sc-24948) and stored in PBS containing 0.1% BSA for on-bead digestion. Liquid chromatography with tandem mass spectrometry was performed by LC at Taplin Biological Mass Spectrometry Facility, Harvard Medical School. Three hearts were used for streptavidin pull-down in each group. Beads were washed at least five times with 100 μ l 50 mM ammonium bicarbonate. Then 5 μ l (200 ng/ μ l) of modified sequencing-grade trypsin (Promega, Madison, WI) was spiked in and the samples were incubated at 37°C overnight. The beads were then removed using a magnet, and the supernatant was dried in a speed-vac. The samples were re-suspended in 50 μ l HPLC solvent A (2.5% acetonitrile, 0.1% formic acid) and desalted by STAGE tip.⁴⁴ On the day of analysis the samples were reconstituted in 10 μ l of HPLC solvent A. A nano-scale reverse-phase HPLC capillary column was created by packing 2.6 μ m C18 spherical silica beads into a fused silica capillary (100 μ m inner diameter x ~30 cm length) with a flame-drawn tip.⁴⁵ After equilibrating the column, each sample was loaded via a Famos autosampler (LC Packings, San Francisco CA). A gradient was formed and peptides were eluted with increasing concentrations of solvent B (97.5% acetonitrile, 0.1% formic acid). As peptides eluted, they were subjected to electrospray ionization and then entered into an LTQ Orbitrap Velos Elite ion-trap mass spectrometer (Thermo Fisher Scientific, Waltham, MA). Peptides were detected, isolated, and fragmented to produce a tandem mass spectrum of specific fragment ions for each peptide. Peptide sequences (and hence protein identity) were determined by matching protein databases with the acquired fragmentation pattern by the software program, Sequest (Thermo Fisher Scientific, Waltham, MA)⁴⁶. All databases include a reversed version of all the sequences and the data was filtered to between a one and two percent peptide false discovery rate.

I. 458 “oC” is missing

Now corrected.

I. 546 “mass spectrometry” should be used instead of “mass spectroscopy”

Now corrected.

REVIEWERS' COMMENTS

Reviewer #1 (Remarks to the Author):

My previous comments have been addressed

Reviewer #2 (Remarks to the Author):

The authors have improved the MS substantially and answered most of our criticisms in a satisfactory manner.

Reviewer #3 (Remarks to the Author):

Thanks to the authors for explanations and the adaptation of the manuscript.

I recommend minor changes in figure 1 E:

- What is the y-axis label “MS signal +1” referring to? I guess displayed is summed or averaged peptide intensity per protein. If so, y-axis should be named with “protein intensity”
- Protein labels on the right hand side are detached from the dots they are referring to. I guess the blue dot on the top belongs to TRDN but the corresponding line points to a dot below.

Reviewer 3:

Thanks to the authors for explanations and the adaptation of the manuscript.

I recommend minor changes in figure 1 E:

- What is the y-axis label "MS signal +1" referring to? I guess displayed is summed or averaged peptide intensity per protein. If so, y-axis should be named with "protein intensity"

We changed the y-axis label to Protein Intensity + 1 as suggested by the reviewer. The axis shows the summed protein intensity plus one.

- Protein labels on the right hand side are detached from the dots they are referring to. I guess the blue dot on the top belongs to TRDN but the corresponding line points to a dot below.

The plot shows three different points (protein intensity signal recovered from Triadin-bioID, Junctin-bioID, and control) for each vertical column (protein rank by ratio of average bioID signal to control signal). Therefore we labeled the vertical column corresponding to selected proteins rather than each point. In fact the top blue dot is not TRDN; TRDN is the third column from the left. For specific values and ranks, we provided Suppl. Data 1.